# Cytoplasm of the Wild Species *Aegilops mutica* Reduces *VRN1* Gene Expression in Early Growth of Cultivated Wheat: Prospects for Using Alloplasmic Lines to Breed Varieties Adapted to Global Warming

**DOI:** 10.3390/plants13233346

**Published:** 2024-11-28

**Authors:** Mina Matsumura, Yuko Watanabe, Hiroko Tada, Koji Murai

**Affiliations:** Graduate School of Bioscience, Fukui Prefectural University, Fukui 910-4103, Japan; c60ssh0r1z0n@gmail.com (M.M.); s2123029@g.fpu.ac.jp (Y.W.); pt-tada@fpu.ac.jp (H.T.)

**Keywords:** *Aegilops mutica*, alloplasmic line, cytoplasmic substitution line, flowering, *Triticum aestivum*, *VRN1*, wheat

## Abstract

In a warm winter due to climate warming, it is necessary to suppress early flowering of autumn-sown wheat plants. Here, we propose the use of cytoplasmic genome effects for this purpose. Alloplasmic lines, or cytoplasmic substitution lines, of bread wheat (*Triticum aestivum*) have cytoplasm from a related wild *Aegilops* species through recurrent backcrossing and exhibit altered characteristics compared with the euplasmic lines from which they are derived. Thus, alloplasmic lines with *Aegilops mutica* cytoplasm show delayed flowering compared with lines carrying normal cytoplasm. In the wheat flowering pathway, *VERNALIZATION 1* (*VRN1*) encodes an APETALA1/FRUITFULL-like MADS box transcription factor that plays a central role in the activation of florigen genes, which induce floral meristems in the shoot apex. Here, we compared expression of *VRN1* alleles in alloplasmic and euplasmic lines after vernalization. We found that alloplasmic wheat showed a lower level of *VRN1* expression after vernalization compared with euplasmic wheat. Thus, nuclear-cytoplasm interactions affect the expression levels of the nuclear *VRN1* gene; these interactions might occur through the pathway termed retrograde signaling. In warm winters, autumn-sown wheat cultivars with spring habit can pass through the reproductive growth phase in very early spring, resulting in a decreased tiller/ear number and reduced yield performance. Here, we present data showing that an alloplasmic line of ‘Fukusayaka’ can avoid the decrease in tiller/ear numbers during warm winters, suggesting that this alloplasmic line may be useful for development of varieties adapted to global warming.

## 1. Introduction

Cytoplasmic substitution (alloplasmic) lines are produced by outcrossing a wheat cultivar with a wild wheat species, followed by recurrent backcrossing. They provide a valuable resource for studying interactions between nuclear and cytoplasmic genomes and for investigating the effects of these interactions on plant phenotypes [1]. Comprehensive studies have examined the effects of cytoplasm from wild wheat species on phenotypic traits in 12 common wheat cultivars/strains using cytoplasmic backcross lines produced from 46 *Aegilops* and *Triticum* species/accessions [2,3]. In normal (euplasmic) lines, the nuclear and cytoplasmic genomes have co-evolved. By contrast, alloplasmic lines have an alien cytoplasm genome that is a mismatch for the nuclear genome. Alloplasmic lines often show phenotypic changes compared with euplasmic lines because of this genomic mismatch [2,3]. One of these phenotypic effects is cytoplasmic male sterility (CMS), which is the most studied nuclear-cytoplasmic (NC) interaction due to its importance in hybrid wheat production [4]. In addition to CMS, NC interactions in alloplasmic wheat lines affect phenotypic aspects, such as vigor and viability [5]. In alloplasmic lines, we believe that these traits are altered because the expression of the nuclear genome is modified by the heterogeneous cytoplasmic genome. Transcriptome analyses of alloplasmic bread wheat carrying *Ae. uniaristata*, *Ae. tauschii,* or *Hordeum chilense* cytoplasm showed that replacement of the wheat cytoplasm with alien cytoplasm altered the transcription patterns of nuclear genes [6]. These changes to nuclear gene expression patterns by NC interactions are associated with the phenotypic changes in the alloplasmic lines, such as germless grain formation, premature sprouting, twin seedling formation, variegation under low temperatures, depressed growth vigor, and delayed heading time [7]. Our group focused on changes in flowering (transition from vegetative to reproductive growth) in alloplasmic lines caused by NC interactions.

In a previous study, we used an alloplasmic line of the bread wheat (*T. aestivum*, 2n = 42, genome formula AABBDD) cultivar ‘Norin 26’ that contains *Ae. mutica* (2n = 14, genome formula MtMt) cytoplasm to develop new alloplasmic lines from 14 Japanese bread wheat cultivars [8]. We then investigated the effects of the *Ae. mutica* cytoplasm on various characteristics in these alloplasmic lines. All of the alloplasmic lines showed delayed heading time (4 to 17 days in the field) compared with the euplasmic lines, and the degree of heading delay depended on the *VERNALIZATION1* (*VRN1*) genotype. In spring wheat cultivars such as ‘Chikugoizumi’, which carry the dominant *VRN-D1* allele, the degree of heading delay due to the alien cytoplasm was large. By contrast, in winter wheat cultivars such as ‘Haruibuki’, which carry recessive *vrn1* alleles in the three homoeologous loci, the heading delay was small. Compared with euplasmic lines, the alloplasmic lines generally showed increased spikelet number per spike but decreased floret number per spikelet, leading to decreased grain number per spike (GNS). However, GNS varied depending on genotype; in the alloplasmic lines of ‘Nanbukomugi’, ‘Nebarigoshi’, and ‘Fukusayaka’, no decrease in GNS occurred. Furthermore, the ‘Nebarigoshi’ and ‘Fukusayaka’ alloplasmic lines did not exhibit the decrease in spike number per plant during winter due to delayed flowering [8].

In the present study, we analyzed plant development and *VRN1* expression patterns in plants of alloplasmic and euplasmic lines grown in a growth chamber. Our analyses indicated that the late-flowering phenotype in the alloplasmic lines is associated with down-regulation of *VRN1* expression in the leaves. To examine whether this suppression of early flowering by *Ae. mutica* cytoplasm is actually useful for warming-adapted wheat breeding, we also examined tiller numbers in alloplasmic and euplasmic ‘Fukusayaka’ plants in the field. During a warm winter, the autumn-sown spring wheat cultivars commenced the reproductive growth phase in very early spring, leading to a decreased tiller/ear number and low yield performance. The ‘Fukusayaka’ alloplasmic lines carrying *Ae. mutica* cytoplasm did not show this decrease in tiller/ear number after the warm winter. These alloplasmic lines should therefore be of use for developing varieties adapted to global warming.

## 2. Results

### 2.1. The Plastochron (Leaf Formation Speed) in Alloplasmic Lines Is Significantly Slower than in Euplasmic Lines

To identify differences in earliness between alloplasmic lines with *Ae. mutica* cytoplasm and euplasmic lines with normal cytoplasm, the alloplasmic lines ‘(mut)-Chikugoizumi’, ‘(mut)-Fukusayaka’, ‘(mut)-Kinuhime’, and ‘(mut)-Haruibuki’ were compared with their corresponding euplasmic lines ‘Chikugoizumi’, ‘Fukusayaka’, ‘Kinuhime’, and ‘Haruibuki’. All the lines were grown in growth chambers under long-day conditions (16 h light/8 h dark, 20 °C). Seedlings at the 1-leaf stage were vernalized for 42 days in a 5 °C growth chamber and then transferred to a 20 °C growth chamber. We measured the time (days) from the 1-leaf stage to flag-leaf unfolding, including the 42-day duration of vernalization in each line (Table 1). The production of successive leaves ceased after the unfolding of the flag-leaf, which could be distinguished from the other leaves by its short blade and the emergence of a spike from its leaf sheath. All four alloplasmic lines showed a consistently delayed timing of flag-leaf unfolding compared to their corresponding euplasmic lines, indicating delayed flowering. The degree of delay was greater in the spring-habit cultivars ‘Chikugoizumi’ (16-day delay) and ‘Fukusayaka’ (14-day delay) than in the winter-habit cultivars ‘Kinuhime’ (10-day delay) and ‘Haruibuki’ (9-day delay).

The leaf stage at the end of the vernalization treatment and the leaf stage at heading are also shown in Table 1. The plastochron (leaf formation speed: time from one leaf unfolding to the next leaf unfolding) for the period from the end of vernalization to the flag-leaf unfolding was calculated for each line (Table 1). In the alloplasmic lines, plastochrons were significantly longer than in the corresponding euplasmic lines. These results indicate that *Ae. mutica* cytoplasm caused a longer plastochron and delayed flowering in alloplasmic lines. Notably, the flag-leaf stage was one to two leaf stages younger in alloplasmic lines than in the euplasmic lines.

### 2.2. VRN1 Expression Levels in Alloplasmic Lines Are Significantly Lower than in Euplasmic Lines

We examined *VRN1* expression in alloplasmic and euplasmic lines after vernalization of the plants in a growth chamber under long-day and normal temperature conditions (16 h light/8 h dark, 20 °C). Bread wheat is hexaploid and has three homoeologous *VRN1* genes located on chromosomes 5A, 5B, and 5D; these genes are named *VRN-A1*, *VRN-B1*, and *VRN-D1,* respectively. The expression pattern of each *VRN1* homoeolog was determined using homoeolog-specific primer sets (Table 2) at different stages of growth (Figs. 1–3). The specificity of the homoeolog-specific primer sets was confirmed by sequencing the PCR products.

All four cultivars tested, ‘Chikugoizumi’, ‘Fukusayaka’, ‘Kinuhime’ and ‘Haruibuki’, carry the recessive *vrn-A1* allele. In the euplasmic cultivars, expression levels increased with growth, peaking at 14 days after the end of vernalization in ‘Chikugoizumi’ and ‘Fukusayaka’ and at 21 days in ‘Kinuhime’ and ‘Haruibuki’ (Figure 1). The peak of *vrn-A1* expression roughly coincided with the period of rapid internode elongation in each cultivar, which indicates the transition from the vegetative to reproductive growth phase. Although the spring wheat cultivars ‘Chikugoizumi’ and ‘Fukusayaka’ have the same allele as the winter wheat cultivars ‘Kinuhime’ and ‘Haruibuki’, they display different expression patterns, suggesting that other genes influence *VRN1* expression (e. g., the dominant *VRN-D1* allele or recessive *vrn-D1* allele). Compared with the euplasmic lines, all four alloplasmic lines showed significantly reduced expression levels.

The B genomes of all four tested cultivars carry the recessive *vrn-B1* allele. The expression patterns of this allele were similar to those observed for *vrn-A1,* except for ‘Haruibuki’ (Figure 2). There was no peak in expression in ‘Haruibuki’, which may be related to the fact that this cultivar is the most winter-hardy of the four tested. Overall, the alloplasmic lines showed significantly reduced expression levels compared to the euplasmic lines.

The spring wheat cultivars ‘Chikugoizumi’ and ‘Fukusayaka’ carry a dominant *VRN-D1* allele, while the winter cultivars ‘Kinuhime’ and ‘Haruibuki’ have a recessive *vrn-D1* allele. In ‘Chikugoizumi’ and ‘Fukusayaka’, *VRN-D1* showed peak expression at 14 days after vernalization, similar to the expression of the recessive alleles of the A and B genomes (Figure 3). The alloplasmic line ‘(mut)-Chikugoizumi’ had significantly reduced expression compared to its euplasmic line; this difference was not present between the alloplasmic and euplasmic lines of ‘Fukusayaka’. It is unclear why expression in ‘(mut)-Fukusayaka’ is not reduced. The expression level of the recessive *vrn-D1* allele in ‘Kinuhime’ and ‘Haruibuki’ was very low in both euplasmic and alloplasmic lines. The fact that expression of the recessive *vrn-D1* allele is largely unaffected by *Ae. mutica* cytoplasm may be the reason for the small degree of delayed flowering in these winter wheat cultivars.

### 2.3. Ae. mutica Cytoplasm Prevents Tiller Number (Ear Number) Reduction During a Warm Winter

To determine how *Ae. mutica* cytoplasm affects tiller/ear number under field conditions, a comparative analysis of ‘Fukusayaka’ was conducted using alloplasmic and euplasmic lines in two seasons: 2021/2022 and 2023/2024 (Figure 4). Temperatures were monitored during these growing seasons, and the average temperatures from early November to late March are shown in Figure 4a. We divided each month into three periods: first, middle, and last, with the average temperatures shown for each period. The 2021/2022 season experienced normal temperatures, with mostly moderate temperature transitions; the lowest temperatures occurred in January and February. The 2023/2024 season, in contrast, showed more drastic temperature fluctuations and a warm period in January and February.

Tiller numbers were counted in each plants of alloplasmic and euplasmic lines in both seasons (Figure 4b). In 2022, the average tiller number was significantly higher in ‘(mut)-Fukusayaka’ plants than in euplasmic ‘Fukusayaka’ plants. In 2024, which experienced a warmer winter, the average tiller number in ‘Fukusayaka’ plants was reduced. The tiller number in ‘(mut)-Fukusayaka’ also decreased significantly in 2024, but was still significantly higher than in ‘Fukusayaka’. Figure 5 shows alloplasmic and euplasmic ‘Fukusayaka’ plants in April 2024, indicating that alloplasmic plants have a sufficient tiller number. These results indicate that *Ae. mutica* cytoplasm is effective in preventing the reduction in tiller/ear number caused by warm weather.

## 3. Discussion

In this study, we demonstrate that alien *Ae. mutica* cytoplasm altered *VRN1* expression levels and patterns in alloplasmic wheat lines, suggesting the occurrence of specific interactions between the cytoplasmic and nuclear genomes (NC interactions). *VRN1* encodes an APETALA1/FRUITFULL-like MADS-box transcription factor that is up-regulated by vernalization [9,10,11,12]. Expression of *VRN1* gradually increases during the seedling growth stage even without vernalization [13], suggesting that the expression of *VRN1* is also controlled by internal signals such as aging. Furthermore, *VRN1* is up-regulated by a long photoperiod [11] and shows a diurnal expression pattern that is affected by the length of daylight [13,14]. These observations indicate that *VRN1* expression is controlled by autonomous and photoperiodic pathways as well as the vernalization pathway. An electrophoretic mobility shift assay using the VRN1 protein expressed as a His-Tag fusion protein in *Escherichia coli* demonstrated that the VRN1 protein directly binds to the CArG-box in the promoter region of the florigen gene *WFT (Wheat FLOWERING LOCUS T*) [15]. This suggests the direct up-regulation of *WFT* by *VRN1*. *WFT* encodes a Raf kinase inhibitor-like protein with a high similarity to the *Arabidopsis* FLOWERING LOCUS T (FT) protein, which is a florigen [16]. In this study, we demonstrated that the expression level of *VRN1* was lower in the alloplasmic line than the euplasmic line after vernalization. Considering that *VRN1* plays a central role in wheat floral development, this reduction in *VRN1* expression level may be related to the delayed flowering of alloplasmic lines.

Genetic information in eukaryotic cells is divided into nuclear and cytoplasmic genomes. Plants contain two cytoplasmic genomes: the mitochondrial genome and the chloroplast genome. Here, the delayed flowering phenotype in alloplasmic wheat lines was identified under normal growth conditions in the field [8] and in a growth chamber using LD conditions after vernalization. We hypothesize that the late heading in the alloplasmic line is caused by retrograde signaling from mitochondria rather than chloroplasts. In contrast to chloroplast genomes, which are highly conserved among plant species, plant mitochondrial genomes are extremely variable in size and structure [17]. Plant mitochondrial genomes are extremely large in comparison to those of animals. They contain considerable amounts of DNA derived from the nuclear and chloroplastic genomes and show a great deal of variation due to recombination involving direct and inverted repeat sequences. Plant mitochondrial genomes have many open reading frames (ORFs) that are suggested to have been produced by inter-molecular recombination during evolution. Some of these novel ORFs are associated with the CMS phenotype in several plant species, such as *ORF129* in petunia and *ORF138* in radish [18]. These associations indicate that mitochondrial ORFs are transcribed, translated, and have a function in the cell. The next step in our research program is to identify genes in the *Ae. mutica* cytoplasmic genome (mitochondrial genome) that affect *VRN1* expression. As *VRN1* is regulated by epigenetic control [19,20,21], candidate genes are likely to be associated with epigenetic regulation. 

In a previous study, we examined the effects of *Ae. mutica* cytoplasm in 14 Japanese wheat cultivars [8]. Alloplasmic lines tended to exhibit longer culm lengths than euplasmic lines. However, *Ae. mutica* cytoplasm had a variable effect on spike number per plant (SNP): in three alloplasmic lines, ‘(mut)-Nebarigoshi’, ‘(mut)-Yumeasahi’, and ‘(mut)-Fukusayaka’, a significant increase in SNP was observed; in two alloplasmic lines, ‘(mut)-Aobakomugi’ and ‘(mut)-Kinuhime’, a significant decrease in SNP was observed. Furthermore, although most alloplasmic lines had lower GNS compared with euplasmic lines due to the significantly reduced number of florets per spikelet, the alloplasmic line of ‘Nebarigoshi’ had a significantly increased GNS, and the alloplasmic lines of ‘Nanbukomugi’ and ‘Fukusayaka’ did not show any change in GNS compared with the euplasmic lines. The differing effects of the *Ae. mutica* cytoplasm on agronomic traits depending on nuclear genotype, indicate a complex NC interaction occurring beyond genes affecting *VRN1* gene expression. 

Currently, we are focusing on ‘(mut)-Fukusayaka’, as this alloplasmic wheat line showed no reduction in GNS. Furthermore, this alloplasmic line has an increased SNP, indicating an acceleration of tillering while reducing internode elongation during the winter season. Early heading is one of the most important agronomic agronomic for bread wheat in East Asia, including Japan, in order to avoid harvesting in the rainy season [22]. Therefore, in central to southwestern Japan, autumn-sown, early-heading spring wheat cultivars that carry *VRN-D1* are cultivated. However, in warm winters, autumn-sown spring wheat cultivars can transit to the reproductive growth phase in very early spring resulting in a decreased tiller/ear number and reduced yield performance. In this study, we demonstrated that the alloplasmic line of ‘Fukusayaka’ was not hampered by a decrease in tiller number during a warm winter. This alloplasmic line should be useful for the development of varieties adapted to global warming. However, in monsoon regions like Japan, wheat varieties that grow as early as possible have been needed to avoid long rains in early summer. Our proposal to suppress winter flowering by using *Ae. mutica* cytoplasm will result in delayed harvest time, so detailed field verification in each region is essential to determine what kind of balance should be maintained between early maturity and winter flowering suppression.

## 4. Materials and Methods

### 4.1. Plant Materials

Four Japanese bread wheat cultivars were used in this study: ‘Chikugoizumi’, ‘Fukusayaka’, ‘Kinuhime’, and ‘Haruibuki’ (Table 1). We used the alloplasmic line of bread wheat cultivar ‘Norin 26’ with *Ae. mutica* cytoplasm developed by Prof. Tsunewaki to produce further alloplasmic lines from these Japanese bread wheat cultivars by recurrent backcrossing [7,8]. All alloplasmic lines used in this study were of the BC_6_ generation.

### 4.2. Growth Chamber Experiments

Four alloplasmic lines and their corresponding euplasmic lines were cultivated in a growth chamber under long-day (16 h light/8 h dark) conditions at 20 °C (light intensity ~100 μE m^−2^ s^−1^). For the vernalization treatment, 1-leaf stage seedlings were transferred to a cold chamber at 5 °C under short-day conditions (10 h light/14 h dark) for 42 days, then returned to long-day conditions in the 20 °C growth chamber. Leaf formation speed (plastochron) and the total number of leaves, including flag-leaf, were examined in three to five plants of each line. Growth stages were defined by leaf-stage: for example, the 4-leaf stage indicates seedlings with four unfolding leaves on the main shoot. Flag-leaf (final leaf) unfolding time was used as an indicator of earliness of heading rather than heading time itself.

### 4.3. Gene Expression Analysis

Expression of *VRN1* in the alloplasmic and euplasmic plants at each leaf stage was analyzed in vernalized plants grown under long-day conditions at 20 °C. Vernalization treatment was performed on 1-leaf seedlings using a growth chamber at 5 °C for 42 days. Seedlings from each line were sampled for leaves beginning to unfold every 7 days from the end of vernalization treatment. Total RNAs were extracted from leaves using ISOGEN (Nippon Gene, Tokyo, Japan); cDNAs were synthesized from the total RNA using an oligo dT primer, in accordance with the protocol for the Ready-To-Go T-primed First-Strand Kit (GE Healthcare Life Sciences, Chicago, IL, USA). Real-time PCR analyses were performed using a LightCycler 2.0 (Roche Diagnostics GmbH, Basel, Switzerland) with the *VRN1* homoeologous gene-specific primer sets (Table 2). The relative quantities of the transcripts were determined using a SYBR Green-labelled amplification product from the gene for the Cell Division Control Protein (*CDCP*) [23], prepared with the primers *CDCP-L* and *CDCP-R* (Table 2). Two biological replicates, with three technical replications, were performed. Each biological sample contained leaves from different individual plants.

### 4.4. Field Experiments

Alloplasmic ‘(mut)-Fukusayaka’ and euplasmic ‘Fukusayaka’ plants were grown in an experimental field at Fukui Prefectural University in two seasons: 2021/2022 and 2023/2024. In each season, daily average temperature data were extracted from the Japan Meteorological Agency homepage. Tiller number was measured on five to seven individual plants of each line; inter-line and inter-year differences were analyzed by an analysis of variance with Fisher’s least significant difference.

## 5. Conclusions

In warm winters, autumn-sown spring wheat cultivars can transit to the reproductive growth phase in very early spring, resulting in a decreased tiller/ear number and reduced yield performance. The alloplasmic line of ‘Fukusayaka’ with *Ae. mutica* cytoplasm should be useful for breeding varieties adapted to global warming, because it can avoid the decrease in tiller/ear number during warm winters by using the effect of NC interaction.

## 6. Patents

The alloplasmic lines described in this paper are not rights-protected by variety registration. Researchers wishing to obtain these materials and use them for research or breeding should contact KM.

## Figures and Tables

**Figure 1 plants-13-03346-f001:**
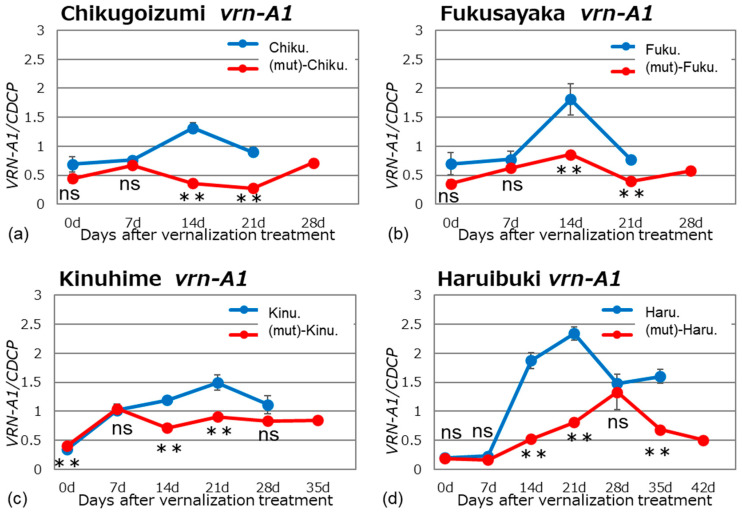
Gene expression patterns of recessive *vrn-A1* in the alloplasmic and euplasmic lines. Plants were grown in a growth chamber under long-day and normal temperature conditions (16 h light/8 h dark, 20 °C) after vernalization treatment: (**a**) ‘Chikugoizumi’, (**b**) ‘Fukusayaka’, (**c**) ‘Kinuhime’, and (**d**) ‘Haruibuki’. Expression levels were measured in leaves that were just unfolding at 7-day intervals from the end of vernalization treatment. The *CDCP* gene was used as an internal control for calculating the relative expression levels of *vrn-A1* genes. Differences between alloplasmic and euplasmic lines were tested for significance using Student’s *t*-tests: ** indicates significant difference at 1%; ‘ns’ indicates no significant difference.

**Figure 2 plants-13-03346-f002:**
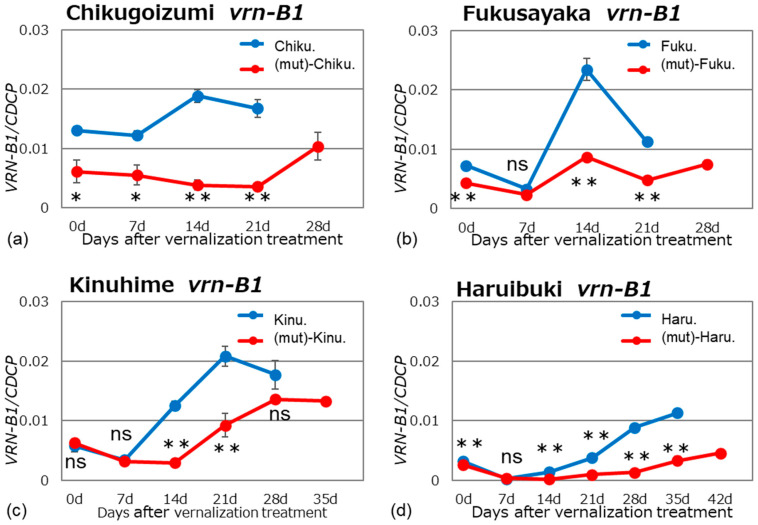
Gene expression patterns of recessive *vrn-B1* in the alloplasmic and euplasmic lines. The experimental conditions were as described in the legend to Figure 1: (**a**) ‘Chikugoizumi’, (**b**) ‘Fukusayaka’, (**c**) ‘Kinuhime’, and (**d**) ‘Haruibuki’. Differences between alloplasmic and euplasmic lines were tested for significance using Student’s *t*-tests: * and ** indicate significant differences at 5% and 1%, respectively; ‘ns’ indicates no significant difference.

**Figure 3 plants-13-03346-f003:**
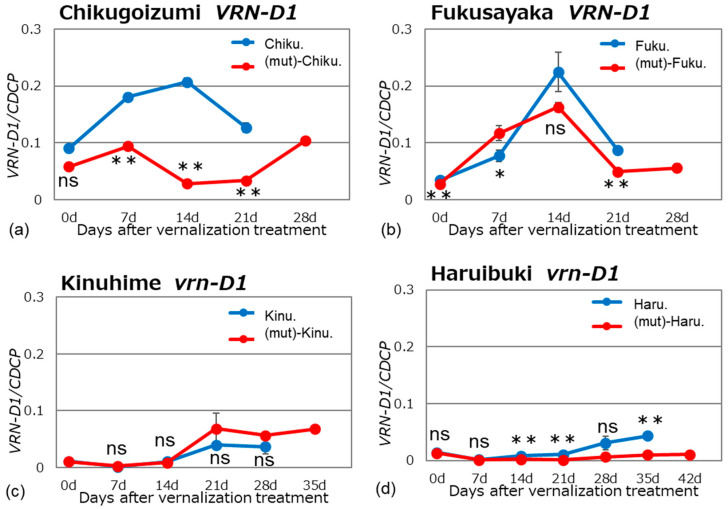
Gene expression patterns of dominant *VRN-D1* and recessive *vrn-D1* in the alloplasmic and euplasmic lines. The experimental conditions were as described in the legend to Figure 1: (**a**) ‘Chikugoizumi’ *VRN-D1*, (**b**) ‘Fukusayaka’ *VRN-D1*, (**c**) ‘Kinuhime’ *vrn-D1* and (**d**) ‘Haruibuki’ *vrn-D1*. Differences between alloplasmic and euplasmic lines were tested using Student’s *t*-tests: * and ** indicate significant differences at 5% and 1%, respectively; ‘ns’ indicates no significant difference.

**Figure 4 plants-13-03346-f004:**
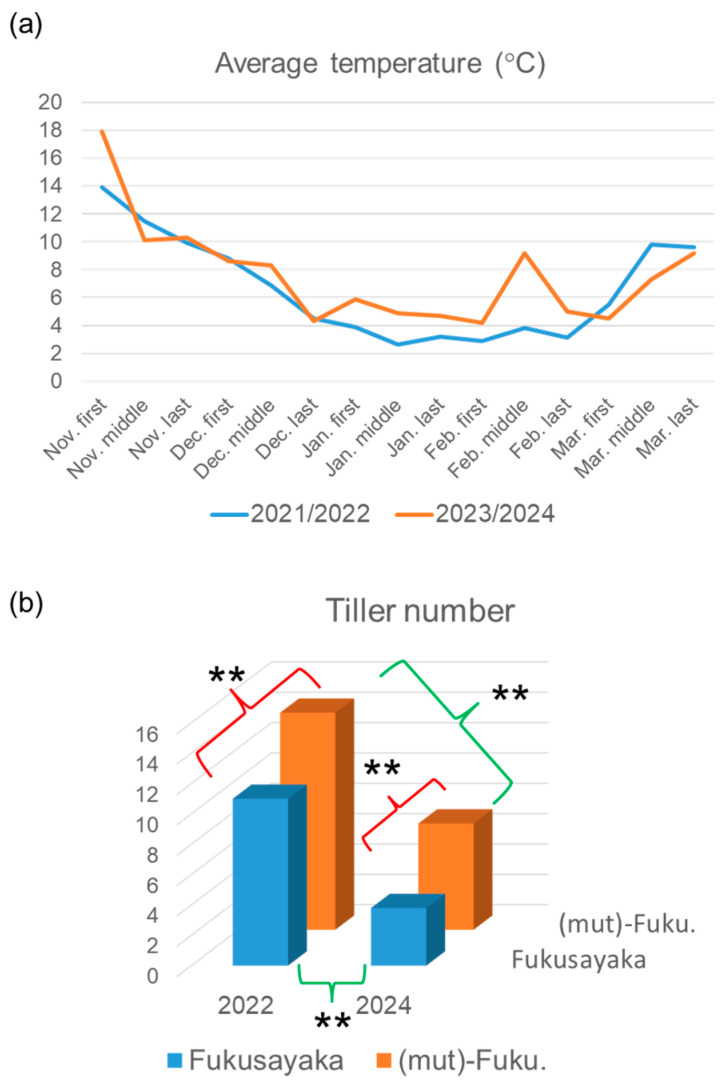
(**a**) Average temperatures from early November to late March. Average temperatures are displayed by dividing each month into three periods: first, middle, and last. The daily average temperature data were extracted from the Japan Meteorological Agency homepage. (**b**) Tiller number of individual alloplasmic ‘(mut)-Fukusayaka’ and euplasmic ‘Fukusayaka’ plants grown in the field in 2021/2022 or 2023/2024 seasons. ** indicates significant difference at 1%.

**Figure 5 plants-13-03346-f005:**
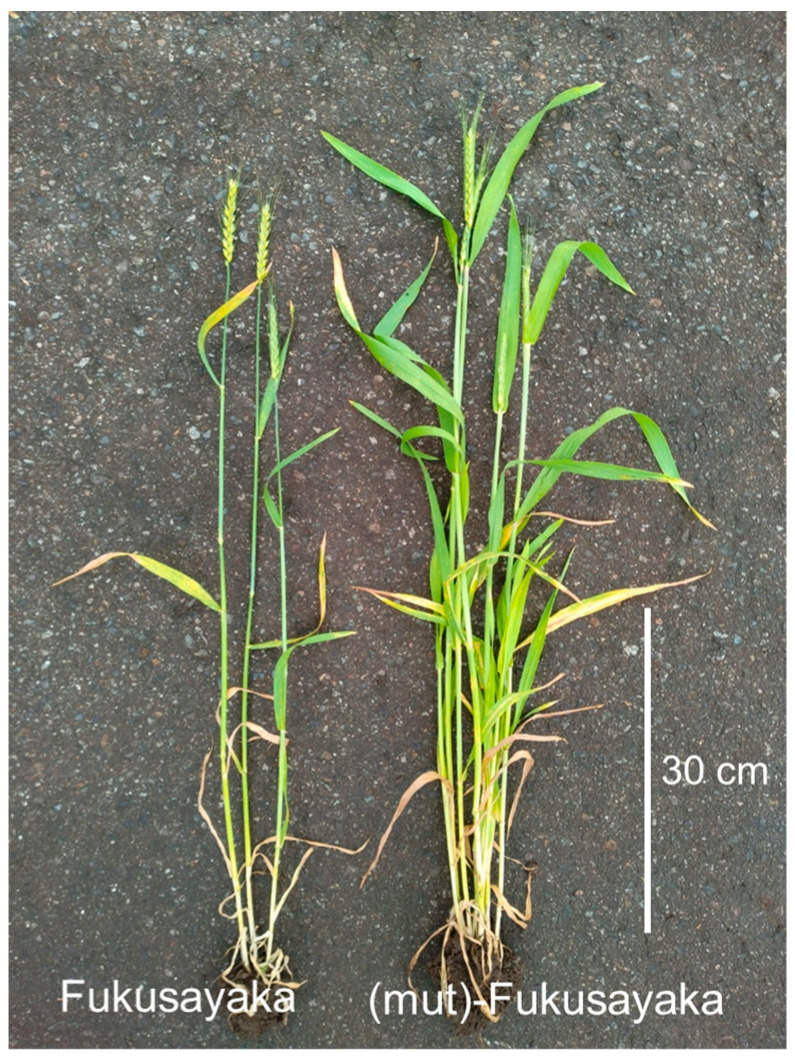
Alloplasmic ‘(mut)-Fukusayaka’ (**right**) and euplasmic ‘Fukusayaka’ plants (**left**) grown in the field in the 2023/2024 season. The image was captured in April 2024.

**Table 1 plants-13-03346-t001:** Plastochron and earliness of alloplasmic and euplasmic lines grown in a growth chamber under long-day conditions (16 h light/8 h dark, 20 °C) after vernalization for 42 days at 5 °C.

Line	Leaf Stage		Plastochron (Days: Mean ± SE)	Days to Flag-Leaf Unfolding ^2^ (Days: Mean ± SE)
	End of Vernalization	Heading		
Chikugoizumi	4.0	8.0	2.25 ± 0.08	65.0 ± 0.3
Fukusayaka	3.8	8.8	3.36 ± 0.33	72.4 ± 0.6
Kinuhime	3.8	9.6	4.13 ± 0.31	77.8 ± 1.0
Haruibuki	3.0	10.0	4.35 ± 0.19	86.2 ± 0.4
(mut)-Chiku.	1.3	7.0	4.38 ± 0.22 **	81.0 ± 0.4 **
(mut)-Fuku.	2.0	7.6	5.48 ± 0.11 **	86.6 ± 0.9 **
(mut)-Kinu.	1.7	8.0	5.02 ± 0.15 *	87.3 ± 0.3 **
(mut)-Haru.	1.2	8.0	5.76 ± 0.15 **	95.2 ± 1.8 **
ANOVA ^1^	-	-	*p* = 0.000 **	*p* = 0.000 **

^1^ An analysis of variance (ANOVA) was first conducted on all lines (** indicates significant difference at 1%), then the least significant difference method was used to detect significant differences between lines. * and ** indicate significant differences between an alloplasmic line and its euplasmic line at 5% and 1%, respectively. ^2^ Including vernalization treatment for 42 days.

**Table 2 plants-13-03346-t002:** Primer sets for gene expression analysis.

Gene	Primer Name	Sequence (5′ to 3′)	Product Size (bp)	Annealing Temperature (°C)
*CDCP*	CDCP-L	CAAATACGCCATCAGGGAGAACATC	227	62
	CDCP-R	CGCTGCCGAAACCACGAGAC		
*VRN-A1*	Real-VRN-A1L	CAGCCTCAAACCAGCTCTTCA	102	65
	Real-VRN-A1R	CTCTGCCCTCTCGCCTGT		
*VRN-B1*	Real-VRN-B1L4	TCGAGAAGCAGAAGGCCCAG	143	65
	Real-VRN-B1R4	CTCTGCCCTCTCTCCTGAT		
*VRN-D1*	Real-VRN-D1L7	ATTCATCCAGCGGCGG	107	65
	Real-VRN-D1R7	CAGCCGTTGATGTGGCTC		

## Data Availability

The original contributions presented in the study are included in the article, further inquiries can be directed to the corresponding author.

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
