# Peer review of "Cytoplasm of the Wild Species Aegilops mutica Reduces VRN1 Gene Expression in Early Growth of Cultivated Wheat: Prospects for Using Alloplasmic Lines to Breed Varieties Adapted to Global Warming"

_plants, 2024, doi:10.3390/plants13233346_

Round 1

Reviewer 1 Report

Comments and Suggestions for Authors

The abstract is just too literature with no figure added. Please present key achievements in the result section.

In the introduction, kindly provide more details on the genome size, chromosome number, previous studies conducted, ploidy, and genomic resources. 

The materials and methods need to be adjusted as the analysis section is weak and needs more details.

How many biological replicates were considered, and how was the sampling performed, what was the time, what was the period, etc.? Present the RNA quality before analysis.

The discussion has to be adjusted based on the new result 

Comments on the Quality of English Language

he abstract is just too literature with no figure added. Please present key achievements in the result section.

In the introduction, kindly provide more details on the genome size, chromosome number, previous studies conducted, ploidy, and genomic resources. 

The materials and methods need to be adjusted as the analysis section is weak and needs more details.

How many biological replicates were considered, and how was the sampling performed, what was the time, what was the period, etc.? Present the RNA quality before analysis.

The discussion has to be adjusted based on the new result 

Author Response

Responses to Reviewer 1

Comments1: The abstract is just too literature with no figure added. Please present key achievements in the result section.

Response 1: Abstract has no figure. The key result of this study is that VRN1 gene expression levels in alloplasmic lines are lower than in euplasmic lines. This result is clearly documented in the Results section.

Comments 2: In the introduction, kindly provide more details on the genome size, chromosome number, previous studies conducted, ploidy, and genomic resources. 

Response 2: I added the information about chromosome number and genome formula in the revised manuscript.

Comments 3: The materials and methods need to be adjusted as the analysis section is weak and needs more details.

Response 3: Materials and Methods section consists of growth chamber experiments, gene expression analysis, field experiments, according to the result section.

Comments 4: How many biological replicates were considered, and how was the sampling performed, what was the time, what was the period, etc.? Present the RNA quality before analysis.

Response 4: I mentioned the detail of methods for gene expression analysis. The quality of the RNA has of course been confirmed, but such details are not usually included in the paper.

Comments 5: The discussion has to be adjusted based on the new result.

Response 5: A new finding of this study is that in alloplasmic lines, cytoplasmic effects reduce VRN1 expression levels early in growth and promote winter tillering. I discussed the points in Discussion section.

Reviewer 2 Report

Comments and Suggestions for Authors

The authors of the paper present a relevant topic, namely the development, based on their study, of wheat varieties adapted to current climate changes, with results that could create a multiplier effect at the level of further research in the field. We suggest highlighting this aspect, particularly regarding the multiplier effect of the authors’ innovative findings, both in the abstract and throughout the chapter presenting the study results. Moreover, we recommend that the authors emphasize, starting from the abstract, the paper’s innovative scientific contributions to the specialized scientific literature.

The authors adequately mention concepts, citations, and bibliographic references, indicating thorough documentation on the research topic. For instance, references such as [6] “Wheat cytoplasm with extraterrestrial cytoplasm altered transcription patterns of nuclear genes.” However, we suggest the authors highlight relevant works in a distinct subchapter, “Literature Review,” in the introductory chapter.

The research methodology is adequately presented in the “Materials and Methods” chapter. Additionally, many of the methods, tools, and analyses used to obtain the study’s results are presented in the results chapter. For example, “a comparative analysis of the Fukusayaka variety was conducted using alloplasmic and euplasmic lines over two seasons, 2021/2022 and 2023/2024.” We also suggest marking this chapter clearly in the study before presenting the results and discussions.

The results are presented descriptively and graphically through tables and figures. Furthermore, the authors demonstrate that “Ae. mutica cytoplasm modified VRN1 expression levels and patterns in alloplasmic wheat lines, suggesting the emergence of specific interactions between cytoplasmic and nuclear genomes (NC interactions),” thus directing the research toward practical application. However, we recommend that the authors highlight in a separate paragraph their innovative scientific contributions to the specialized literature.

The conclusions are presented succinctly, with references to the study’s practical implications, such as “in warm winters, spring wheat varieties sown in autumn may transition to the reproductive growth phase very early in spring, thus reducing yield performance.” Furthermore, we suggest that the authors highlight the study's limitations and present future research directions.

We congratulate the research team on the studied topic and recommend revising the paper according to the above suggestions.

Comments on the Quality of English Language

I did not identify any major language deficiencies!

Author Response

Comments 1: The authors of the paper present a relevant topic, namely the development, based on their study, of wheat varieties adapted to current climate changes, with results that could create a multiplier effect at the level of further research in the field. We suggest highlighting this aspect, particularly regarding the multiplier effect of the authors’ innovative findings, both in the abstract and throughout the chapter presenting the study results. Moreover, we recommend that the authors emphasize, starting from the abstract, the paper’s innovative scientific contributions to the specialized scientific literature.

Response 1: I added the sentences in the beginning of Abstract to emphasize our findings.

Comments 2: The authors adequately mention concepts, citations, and bibliographic references, indicating thorough documentation on the research topic. For instance, references such as [6] “Wheat cytoplasm with extraterrestrial cytoplasm altered transcription patterns of nuclear genes.” However, we suggest the authors highlight relevant works in a distinct subchapter, “Literature Review,” in the introductory chapter.

Response 2: I have added a sentence to the Introduction section to clearly identify the literature review on gene expression in alloplasmic lines.

Comments 3: The research methodology is adequately presented in the “Materials and Methods” chapter. Additionally, many of the methods, tools, and analyses used to obtain the study’s results are presented in the results chapter. For example, “a comparative analysis of the Fukusayaka variety was conducted using alloplasmic and euplasmic lines over two seasons, 2021/2022 and 2023/2024.” We also suggest marking this chapter clearly in the study before presenting the results and discussions.

Response 3: Thank you for pointing this out. However, I don't think it is advisable to rearrange the order of the results as the plot of the paper. Then, I have added a sentence in the Introduction about the Fukuyasaya.

Comments 4: The results are presented descriptively and graphically through tables and figures. Furthermore, the authors demonstrate that “Ae. mutica cytoplasm modified VRN1 expression levels and patterns in alloplasmic wheat lines, suggesting the emergence of specific interactions between cytoplasmic and nuclear genomes (NC interactions),” thus directing the research toward practical application. However, we recommend that the authors highlight in a separate paragraph their innovative scientific contributions to the specialized literature.

Response 4: I have highlighted our results by adding sentences to the Abstract and Introduction.

Comments 5: The conclusions are presented succinctly, with references to the study’s practical implications, such as “in warm winters, spring wheat varieties sown in autumn may transition to the reproductive growth phase very early in spring, thus reducing yield performance.” Furthermore, we suggest that the authors highlight the study's limitations and present future research directions.

Response 5: In the end of Discussion section, I added sentences to explain the limitations of use of Ae. mutica cytoplasm.

Round 2

Reviewer 2 Report

Comments and Suggestions for Authors

The authors of the paper present a relevant topic, but as I have specified before, it is very important that with the revision process they reduce the similarity index to below 10%, which is why we suggest to the authors, based on the similarity report, the complete revision of the paper. Thank you!

Author Response

Comments:

The authors of the paper present a relevant topic, but as I have specified before, it is very important that with the revision process they reduce the similarity index to below 10%, which is why we suggest to the authors, based on the similarity report, the complete revision of the paper. Thank you!

Reply:

In the revised manuscript, we have addressed the reviewers' comments one by one. In addition, some minor revisions were made to the final manuscript to make it easier for readers to understand. Our previous paper on the breeding of cytoplasmic replacement lines and their agronomic traits has similarity in the material, but the content is quite different.